# Ly6C^hi^ Monocytes Are Metabolically Reprogrammed in the Blood during Inflammatory Stimulation and Require Intact OxPhos for Chemotaxis and Monocyte to Macrophage Differentiation

**DOI:** 10.3390/cells13110916

**Published:** 2024-05-26

**Authors:** Gareth S. D. Purvis, Eileen McNeill, Benjamin Wright, Keith M. Channon, David R. Greaves

**Affiliations:** 1Sir William Dunn School of Pathology, University of Oxford, Oxford OX1 3RE, UK; gareth.purvis@cardiov.ox.ac.uk; 2Wellcome Trust Centre for Human Genetics, University of Oxford, Oxford OX3 7BN, UK; eilsmcneill@gmail.com (E.M.); benjamin.wright@well.ox.ac.uk (B.W.); keith.channon@cardiov.ox.ac.uk (K.M.C.); 3Division of Cardiovascular Medicine, Radcliffe Department of Medicine, University of Oxford, Oxford OX3 9DU, UK; 4British Heart Foundation Centre of Research Excellence, University of Oxford, Oxford OX3 9DU, UK

**Keywords:** Ly6C^hi^ monocytes, single cell transcriptomics, OxPhos, chemotaxis

## Abstract

Acute inflammation is a rapid and dynamic process involving the recruitment and activation of multiple cell types in a coordinated and precise manner. Here, we investigate the origin and transcriptional reprogramming of monocytes using a model of acute inflammation, zymosan-induced peritonitis. Monocyte trafficking and adoptive transfer experiments confirmed that monocytes undergo rapid phenotypic change as they exit the blood and give rise to monocyte-derived macrophages that persist during the resolution of inflammation. Single-cell transcriptomics revealed significant heterogeneity within the surface marker-defined CD11b^+^Ly6G^−^Ly6C^hi^ monocyte populations within the blood and at the site of inflammation. We show that two major transcriptional reprogramming events occur during the initial six hours of Ly6C^hi^ monocyte mobilisation, one in the blood priming monocytes for migration and a second at the site of inflammation. Pathway analysis revealed an important role for oxidative phosphorylation (OxPhos) during both these reprogramming events. Experimentally, we demonstrate that OxPhos via the intact mitochondrial electron transport chain is essential for murine and human monocyte chemotaxis. Moreover, OxPhos is needed for monocyte-to-macrophage differentiation and macrophage M(IL-4) polarisation. These new findings from transcriptional profiling open up the possibility that shifting monocyte metabolic capacity towards OxPhos could facilitate enhanced macrophage M2-like polarisation to aid inflammation resolution and tissue repair.

## 1. Introduction

As an acute inflammatory response evolves, the cellular composition at the site of inflammation rapidly changes [1]. The initial recruitment of polymorphonuclear leukocytes (PMNs) is overtaken by an influx of monocytes from the circulation. Monocytes are divided into three main classes in humans and mice: classical (CD14^+^CD16^−^ in humans and Ly6C^hi^ in mice), intermediate (CD14^+^CD16^+^ in humans and Ly6C^+^Treml4^+^ in mice), and nonclassical (CD14^−^CD16^+^ in humans and Ly6C^lo^ in mice) [2]. During the inflammatory response, monocytes are released into the blood from two main stores: the bone marrow, which is the origin of all haemopoietic lineages, and the spleen [3,4,5]. This process is tightly regulated through the release of specific chemoattractants and the expression of their cognate receptors on target cells [6].

Once monocytes are released from either the bone marrow or spleen, chemokine/chemokine receptor interactions lead blood monocytes to the site of inflammation. To date, little is known about how metabolic demand affects monocyte chemotaxis, while work in T-cell migration has shown a central role for glycolysis [7]. Following peritoneal challenge with zymosan, in mice, intermediate Ly6C^+^Treml4^+^ and classical Ly6C^hi^ monocytes are rapidly recruited to the inflamed peritoneum. Our current understanding suggests that monocytes utilise glycolytic metabolism after activation; however, most studies measure this ex vivo and using lipopolysaccharide as the inflammatory stimulus [8,9]. Non-classical Ly6C^lo^ monocytes tend to have a patrolling function and show less recruitment to the peritoneum [10,11], but little is known about their metabolic profile in vivo. Monocytes that are recruited to a site of inflammation can undergo differentiation into dendritic cells or macrophages.

In the latter stages of the peritoneal inflammatory response, macrophages are the predominant cell type present. However, the exact origin of ‘resolution-phase’ macrophage populations has been difficult to pin down and their function is less commonly studied [12]. The classical paradigm of M1 (pro-inflammatory) and M2 (pro-resolution) macrophage differentiation is currently being challenged, with limited evidence of distinct classes of macrophages described in vitro being present in vivo [13]. It is likely that macrophage differentiation in vivo is more dynamic and complex than macrophage differentiation in vitro [14]. Tissue-resident macrophage populations are classically thought of as ‘M2-like’, as they perform homoeostatic functions [15].

Monocyte-derived macrophages recruited following inflammatory stimulus exhibit distinct transcriptional, functional, and phenotypic signatures compared to resident macrophages. In the peritoneal cavity, sterile inflammation causes substantial loss in the number of peritoneal macrophages through egress, cell death, or the loss of fibrin clots [16], but the extent of this loss appears to be dependent on the stimulus and the severity of the ensuing inflammation [17]. The peritoneal macrophage population can be replenished through proliferation during the resolution phase [18]. During peritoneal inflammation, there can be loss of resident cells, which are partially replaced by bone marrow-derived cells; again, the degree of replacement correlates with the extent of initial loss [19,20].

It is becoming ever clearer that monocytes and macrophages have considerably more heterogeneity than can be determined by cluster differentiation (CD) expression alone [21,22,23]. Plasticity within the system has been a stalwart of the innate immune response [24,25]. However, this many not be as prevalent as once thought; indeed, work by Rhee et al. suggests that switching between monocyte subclasses is uncommon even under extreme physiological challenges [26]. Meanwhile, terminally differentiated macrophages have limited class switching plasticity, unless the metabolic re-wiring that drives M1 polarization is blocked by inhibiting nitric oxide production [27].

In this study, we sought to investigate if there is heterogeneity within circulating and recruited Ly6C^hi^ monocyte populations and to identify the earliest pathways that regulate monocyte recruitment and differentiation at sites of zymosan-induced peritonitis. Single-cell transcriptomics allowed us to compare the transcriptome of monocytes in the blood (naïve to inflammatory stimuli), with newly recruited monocytes (in the peritoneum post inflammatory stimuli), without the need for tissue digestion and lengthy sorting protocols. This experimental design provided unique insights into how circulating blood Ly6C^hi^ monocytes are reprogrammed during the earliest events of acute resolving inflammation in a spaciotemporal manner at a single-cell resolution. Our study revealed (1) that there is previously underappreciated heterogeneity within Ly6C^hi^ monocyte population in the blood; (2) that Ly6C^hi^ monocytes’ cellular fate at a site of inflammation is pre-determined in the blood; (3) that a sub-set of Ly6C^hi^ monocytes which are pre-programmed to become M2-like macrophages are dependent on a metabolic switch towards oxidative phosphorylation as early as 2 h post inflammatory stimulation in the blood; and (4) that OxPhos via the intact mitochondrial electron transport chain is critical for monocyte chemotaxis, monocyte-to-macrophage differentiation and M2/M(IL-4) polarisation.

## 2. Methods

### 2.1. Animal Ethics

All animal studies were conducted with ethical approval from the Local Ethical Review Committee at the University of Oxford and in accordance with the UK Home Office regulations (Guidance on the Operation of Animals, Scientific Procedures Act, 1986). Mice were housed in ventilated cages with a 12 h light/dark cycle and controlled temperature (20–22 °C) and fed normal chow and water ad libitum. For in vivo experiments, male adult mice > 10 weeks of age were used for all studies. Mice had received no prior procedures (including acting as breeding stock) prior to use in experiments for this manuscript.

### 2.2. Zymosan-Induced Peritonitis

Male mice were injected with 100 µg of zymosan A (Sigma) in 500 µL of PBS using an 30 G insulin syringe. At defined time points, mice underwent peritoneal lavage post-mortem with ice-cold PBS/EDTA (5 mM), lavage fluid was injected using a 25 G needle the cavity was mixed via agitation, and then lavage fluid was withdrawn using a 21 G needle. Cellular recruitment was assessed by means of flow cytometry using an absolute cell count method as previously described [28]. Total cellular recruitment was calculated as the total number of cells that would be recovered in the 5 mL lavage volume.

### 2.3. Adoptive Transfer

*Protocol A:* Concurrently, hCD68-GFP and C57BL/6J mice were administered 100 µg of zymosan A in PBS (Sigma, St. Louis, MO, USA) i.p. After 2 h, the hCD68-GFP mice were sacrificed and peritoneal exudates were collected via peritoneal lavage in sterile ice-cold PBS/EDTA (5 mM), counted and washed twice in PBS. A total of 1.5 × 10^6^ cells were then injected intraperitoneally (i.p.) into the C57BL/6 mice. After a further 20 or 44 h, the recipient mice were sacrificed and peritoneal exudates were collected via peritoneal lavage in ice-cold PBS/EDTA (5 mM). Total cell counts and the cellular composition of peritoneal exudate were determined via flow cytometry.

*Protocol B:* Monocytes were isolated from hCD68-GFP bone marrow as described previously [29]. Briefly, bone marrow cell suspensions were passed through a 70 µm cell strainer and red blood cells were lysed (LCK Buffer, Sigma) for 10 min on ice. The bone marrow cell suspension were isolated using the Murine Bone marrow monocyte isolation kit (Miltenyi Biotec, Bergisch Gladbach, Germany) and underwent negative selection using an autoMACS Pro Separator (Miltenyi Biotec). The purity of the resulting populations was confirmed by means of flow cytometry. Bone marrow isolations from a total of 2 femurs typically yielded 2 × 10^6^ cells at a purity of 90% monocytes.

Isolated h*CD68*-GFP monocytes (1 × 10^6^, in 200 µL) were delivered intravenously 30 min after C57BL/6J mice were administered 100 µg of zymosan A in PBS (Sigma) i.p. After 24 or 48 h mice were sacrificed and peritoneal exudates were collected by means of peritoneal lavage in ice-cold PBS/EDTA (5 mM). Total cell counts and the cellular composition of peritoneal exudate were determined by means of the flow cytometry full gating strategy found in Appendix A.

### 2.4. Cell Capture, cDNA Synthesis, Single-Cell RNA-Seq Library Preparation

Whole blood and peritoneal exudate cells were isolated from mice. Red blood cells were lysed (LCK Buffer, Sigma) for 10 min on ice and cells were stained for flow cytometry. Briefly, cells were live/dead stained using IR Live Dead Stain, Thermofisher (Waltham, MA, USA). Cells were incubated with Fc-blocking anti-CD16/32 antibodies followed by cell hashing antibody (1 µg/mL, Cell Hashing A, BioLegend, San Diago, CA, USA) combined and stained with anti-CD11b-PerCP, Ly6G-FITC, and Ly6C-PE antibodies (BioLegend, San Diago, CA, USA). The cell hashing approach allowed each of the 5 biological replicates from the 4 experimental groups to be combined and sorted into one sample per experimental group, whilst still capturing data from each individual mouse.

Stained cells were then sorted and CD11b^+^LygG^−^Ly6C^+^ monocytes were collected and washed in PBS with 0.04% BSA and re-suspended at a concentration of ~50 cells/µL before capturing single cells in droplets on the Chromium 10x Genomics platform. Library generation for 10x Genomics v3.0 chemistry was performed following the Chromium Single-Cell 3ʹ Reagents Kits User Guide: v3.0 rev A, CG000183 (10x Genomics, Pleasanton, CA, USA). Quantification of cDNA was performed using Qubit dsDNA HS Assay Kit (Life Technologies, Q32851, Carlsbad, CA, USA) and Agilent High-Sensitivity D5000 ScreenTape (Agilent, 5067-5592, Santa Clara, CA, USA). Quantification of the library was performed using the Qubit dsDNA HS Assay Kit (Life Technologies, Q32851) and D1000 ScreenTape (Agilent, 5067-5582). Libraries were prepared and sequenced, with an average of 38,628 ± 5029 reads per cell.

### 2.5. Sequencing and Base Calling

Both the 10× Single-Cell RNAseq 3′ v3 and the Cell Hashing libraries were sequenced together on a single Illumina HiSeq 4000 lane at the Oxford Genomics Centre using a 28 × 98 bp paired-end read configuration. Base-calling and demultiplexing were performed using the “cellranger mkfastq” pipeline from the 10x Genomics Cell Ranger Suite (v3.0.2) (10x Genomics) in order to produce Cell Ranger compatible FastQ files.

### 2.6. Quality Control and Processing

FastQ files were quality controlled using FastQC [30]. Gene expression and cell hashing libraries originating from the same sample were processed together using the “cellranger count” pipeline with the “—libraries” option and default arguments. Data were aligned against the pre-built mouse reference available from 10X Genomics (10x Genomics) which uses GRCm38 (Ensembl 93) primary assembly and annotation files. Cell Ranger’s “count” pipeline performed the alignment, filtering, barcode counting, and UMI counting processes. It used the Chromium cellular barcodes to generate a feature-barcode matrix, and because both the gene expression and cell hashing libraries were provided together, the matrix included both the gene expression and cell hashing feature counts. Cell hashing efficiency was assessed with ridge plots analysis and 5 distinct signatures detected allowing for biological variation within experimental group to be factored into all subsequent analysis (Appendix A).

### 2.7. Single-Cell Analyses

The previously generated gene expression and cell hashing counts matrix was parsed using Seurat v3 [31]. Single cells were captured in the 10× chromium. Initial QC included exclusion of doublets and cells with aberrant ribosomal RNA to genomic RNA. Cells were assigned to clusters using Seurat and the expression profiles for the clusters were examined to determine if any were cells of an unexpected type. Such cells were excluded from the dataset and the clusters were recalculated. The re-clustered data were taken forward for subsequent analysis. Characteristic markers for the new clusters were also established using Seurat.

#### 2.7.1. Pseudotime Analysis

The Monocle3 R package [32,33,34,35] was used to estimate a pseudotime progression for the cells. Initial analysis produced disjointed pseudotime progressions across the different clusters found. The dataset was then subdivided iteratively until each pseudotime model showed one progression for all cells in that set. Each pseudotime model required the specification of a set of ‘root’ cells assumed to be the start of the pseudotime progression. This was taken to be those cells in the model belonging to the naive blood group, if any were present. If not, then the blood 2 h cells were used. If no cells belonging to either group were present, PEC 2hr was taken as the set of root cells (this was for pseudotime models I–VIII). Alternative models used the cluster analysis to establish the set of root cells for further refinement (models IX–XI).

#### 2.7.2. BioGel Elicitation of Primary Mouse Macrophages

Adult male C57BL/6 mice were injected i.p. with 1 mL of sterile 2% bio-gel polyacrylamide beads (P100 fine 45–90 µm) suspended in PBS. Mice were sacrificed 4 days later, and the elicited cells were collected by means of peritoneal lavage with 10 mL of ice-cold PBS/2 mM EDTA. Cells were passed through a 40 µM strained and pelleted by centrifugation (300 G; 5 min).

### 2.8. Generation of Murine Bone Marrow-Derived Macrophages (BMDMs)

Bone marrow was obtained by flushing the femur and tibia of adult female mice with PBS. A single-cell suspension was prepared by passing the bone marrow through a 70 µm cell strainer. Whole bone marrow cultures were then cultured in 10 cm non-tissue-culture-treated dishes for 6 or 7 days in Dulbecco’s modified Eagle’s medium (DMEM) containing 25 mM glucose (Invitrogen, Waltham, MA, USA) and supplemented with 100 U/mL penicillin, 100 ng/mL streptomycin (Sigma), 10% (*v*/*v*) foetal bovine serum, and 10% (*v*/*v*) L929 (ATCC NCTC clone 929) conditioned medium at 37 °C and 5% CO_2_. For polarisation studies, BMDMs were scrapped, counted, and reseeded at an appropriate density in DMEM containing 25 mM glucose (Invitrogen) and supplemented with 100 U/mL penicillin, 100 ng/mL streptomycin (Sigma), and 2% (*v*/*v*) foetal bovine serum. For chemotaxis, on day 7, BMDMs were scrapped counted and resuspended at a density of 2 × 10^6^ per mL.

For monocyte-to-macrophage differentiation experiments, bone marrow monocytes were immuno-magnetically isolated via negative selection and cultured for 7 days in DMEM containing 25 mM glucose (Invitrogen) and supplemented with 100 U/mL penicillin, 100 ng/mL streptomycin (Sigma), 10% (*v*/*v*) fetal bovine serum (PAA Laboratories, Cölbe, Germany), and 10% (*v*/*v*) L929 (ATCC NCTC clone 929) conditioned medium at 37 °C and 5% CO_2_. The protocol for macrophage culture was validated by flow cytometry analysis of the differentiated cells using F4/80 and CD11b as macrophage cell markers.

### 2.9. Isolation and Preparation of Human Monocyte Derived Monocytes (hMoDMs)

Peripheral blood mononuclear cells (PBMC) were purified from leukocyte cones from healthy volunteers with informed consent (NHSBTS, Oxford, UK) by means of density centrifugation over Ficoll-Paque PLUS (Sigma). The PBMC layer was carefully harvested and then washed twice with PBS. Monocytes were isolated from the PBMCs by means of negative selection using magnetic beads (Miltenyi Biotec, Bergisch Gladbach, Germany). Cells were maintained in RPMI 1640 medium supplemented with 1% human serum, 50 ng/mL macrophage colony stimulating factor (hM-CSF, BioLegend, San Diego, CA, USA), 100 U/mL penicillin, and 100 ng/mL streptomycin (Sigma) for 7 days.

### 2.10. Flow Cytometry

Cells were washed in FACS buffer (0.05% BSA, 2 mM EDTA in PBS pH 7.4) blocked using anti CD16/32 for 10 min at 4 °C, followed by antibody staining for the surface markers. The gating strategy for assessing the myeloid cell composition of peritoneal exudates can be found in Appendix A. Absolute cell counts were obtained using a defined quantity of calibration beads added to each sample (CountBright, Invitrogen). Data were acquired using either a DAKO CyAn cytometer and Summit software (both Beckton Coulter, Brea, CA, USA) or a BD Fortessa X20 cytometer and Diva software (both BD Biosciences, Franklin Lakes, NJ, USA) and then analysed using FlowJo v10 (Tree Star Inc., Ashland, OR, USA) software.

### 2.11. Seahorse XFe96 Analysis of Mitochondrial Function

On day 6 of differentiation, BMDMs were plated at a density of 7.5 × 10^4^ cells per well into XF^e^96 microplates. Cells were left to attach for 2 h before stimulation with IL-4 for M2/M(IL-4) polarisation, LPS (100 ng/mL)/IFNγ (20 ng/mL) for M1/M(LPS/IFNγ) polarisation, or left unstimulated (deemed as M0) for 18 h. After 17 h, the medium was changed to DMEM without phenol containing 10 mM glucose, 2 mM L-glutamine, and 2 mM sodium pyruvate. After 1 h, six baseline oxygen consumption rate (OCR) measurements were taken, followed by three measurements after sequential injection of the following compounds: oligomycin (1 µM, injection port A), FCCP (1.5 µM, injection port B), and a combination of antimycin A (1 µM) and rotenone (1 µM) (injection port C). Cells were measured in 4 replicate wells and all Seahorse data were expressed as OCR per 7.5 × 10^5^ cells [36].

Human monocytes were seeded at a density of 1 × 10^5^ per well and differentiated in RPMI 1640 medium supplemented with 1% human serum, 50 ng/mL macrophage colony stimulating factor (hM-CSF, BioLegend, San Diago, CA, USA), 100 U/mL penicillin, and 100 ng/mL streptomycin (Sigma) for 6 days. Eighteen hours before running the MitoStress assay, cells were stimulated with IL-4 for M2/M(IL-4) polarisation, LPS (100 ng/mL)/IFNγ (20 ng/mL) for M1/M(LPS/IFNγ) polarisation, or left unstimulated (deemed as M0).

For calculations, the last reading in each case was used as the most stable point, with the exception of the highest reading following the addition of FCCP. Basal OCR = Base-line − Rotenone/antimycin A; ATP-linked OCR = Base-line − Oligomycin; Maximal OCR = FCCP − Rotenone/antimycin A; Spare capacity = Maximal OCR − Base-line OCR and Proton leak = Base-line OCR − ATP-linked OCR. Data were gathered on Seahorse XFe96 Analyzer (Agilent Technologies, Santa Clara, CA, USA) and analysed using Wave Software (Agilent Technologies).

### 2.12. ACEA xCELLigence Real-Time Cell Migration

Experiments were carried out with CIM-16 plates and an xCELLigence RTCA-DP instrument (ACEA, San Diego, USA) as previously described [37]. Briefly, chemoattractants were made to desired concentrations (C5a, 10 nM, CCL2 10 nM) and loaded into the lower wells of the CIM-16 plate. Upper wells were filled with chemotaxis buffer and plates were equilibrated for 30 min at RT. Freshly isolated murine BioGel-elicited monocytes/macrophages, human monocytes, or BMDMs were resuspended in chemotaxis buffer and incubated with OxPhos inhibitors 15 min at 37 °C and 5% CO_2_. Cell suspensions were placed into the wells of the upper chamber, and the assay was performed over 6 h (readings were taken every 15 s).

### 2.13. Materials

All reagents were purchased from Sigma-Merck unless otherwise stated. A full list of reagents is available in Appendix B Table A1.

### 2.14. Statistical Analysis

When the mean of two experimental groups was compared, a two-tailed Student’s *t*-test was performed. Normally distributed data without repeated measurements were assessed by means of one-way ANOVA followed by Bonferroni correction if the *F* value reached significance at the *p*  <  0.05 level, and there was no significant variance between groups. In all cases, a *p*  <  0.05 was deemed significant.

## 3. Results

### 3.1. Resolution Phase Macrophages Derive from Blood Monocytes

We used low-dose zymosan to induce peritonitis in mice, as a model of acute self-resolving inflammation characterised by a rapid infiltration of myeloid cells into the peritoneum, which is largely resolved within 48 h [38]. After the injection of zymosan, infiltrating PMNs (neutrophils: CD11b^+^Ly6C^+^Ly6G^+^; and then monocytes: CD11b^+^Ly6C^+^Ly6G^−^) populate the peritoneum within 4 h (Figure 1A). Importantly in this model, we observe rapid loss of the resident macrophage (CD11b^+^F4/80^+^CD115^+^) population after zymosan challenge, while after resolution, a new macrophage population is seen (CD11b^+^CD115^+^F4/80^lo^Ly6C^−^) (Figure 1B). Using hCD68-GFP as a marker of the macrophage lineage, we saw a recruitment of monocytes in zymosan-challenged mice which was followed by a sequential appearance of hCD68-GFP^+^ cells with a differing macrophage cell surface marker expression to that of resident peritoneal macrophages (Figure 1C). Given the sequential nature of the appearance of these cells, we hypothesised that these cells may be differentiated from the recruited monocyte population and give rise to the new macrophage population (CD11b^+^CD115^+^Ly6C^−^).

To test the hypothesis that the CD11b^+^CD115^+^Ly6C^−^ cells found in the peritoneal cavity 2 h post zymosan were monocyte-derived, we adoptively transferred hCD68-GFP^+^ cells present in the peritoneum lavage harvested 2 h after zymosan challenge into mice with ongoing peritonitis (Appendix A). Critically, at this point, resident peritoneal macrophages are no longer present in the lavage population and the cell population largely consists of recruited monocytes and neutrophils (Appendix A). Later, 24 and 48 h post cell transfer, we isolated total peritoneal cells and assessed cell surface expression markers. We saw that hCD68-GFP^+^ cells were present in peritoneal exudates 24 h and 48 h after adoptive transfer (Appendix A); additionally, adoptively transferred hCD68-GFP^+^ cells underwent the same differentiation pattern as hCD68-GFP^−^ cells from recipient mice within the peritoneum (CD11b^+^Ly6C^+^CD115^−^ to CD11b^+^Ly6C^−^CD115^+^) (Appendix A). These observations indicate that the new population of CD11b^+^Ly6C^−^CD115^+^ macrophages present 48 h after zymosan injection are derived from cells present in the peritoneum 2 h post zymosan injection, although their origin remains uncertain. We reasoned that they may arise from either local proliferation of a progenitor cell population present in the peritoneum, or most likely, from infiltrating cells.

We excluded the alternative possibility that the peritoneal macrophage populations seen at 24 and 48 h post injection arose from a rare progenitor cell population. We harvested bone marrow monocytes from hCD68-GFP^+^ mice and adoptively transferred these intravenously into zymosan-challenged hCD68-GFP^−^ recipient mice. We have previously shown that bone marrow monocytes undergoing the intravenous adoptive transfer from hCD68-GFP^+^ into recipient WT mice are recruited into the peritoneum following zymosan injection). In the present experiment, we tracked these cells for 24 and 48 h and assessed whether they differentiated into monocyte-derived macrophages (Figure 1D). Bone marrow monocytes from hCD68-GFP^+^ mice were isolated via negative selection and adoptively transferred (i.v.) into recipient WT mice 30 min after zymosan injection. hCD68-GFP^+^ monocytes were present in peritoneal exudates 24 h and 48 h after adoptive transfer (Figure 1E). Adoptively transferred hCD68-GFP^+^ cells underwent the same differentiation pattern as hCD68-GFP^−^ recipient mice cells within the peritoneum (CD11b^+^Ly6C^+^CD115^−^ to CD11b^+^Ly6C^−^CD115^+^) following zymosan challenge (Figure 1F). We therefore concluded from these experiments that CD11b^+^Ly6C^−^CD115^+^ cells present 24/48 h after zymosan injection differentiate from recruited blood monocytes. However, from these experiments, we cannot conclude where (in the blood or at the site of inflammation) monocytes receive the endogenous signalling that initiates the reprogramming needed to terminally differentiate into a monocyte-macrophages at the site of inflammation.

### 3.2. Single-Cell Transcriptomics Reveals Heterogeneity within Ly6C^hi^ Monocyte Populations

To capture the earliest transcriptional reprogramming of circulating and recruited Ly6C^hi^ monocytes, we used a single-cell RNA sequencing approach. We chose three timepoints and isolated either circulating or peritoneal exudate cell (PEC) monocytes: 0 h (naïve blood); 2 h (blood and PEC 2 h); and 6 h (PEC 6 h) after zymosan challenge, as at these time points there was monocyte recruitment to the peritoneum, but this is prior to known alterations in their cell surface markers. This allowed fluorescent antigen cell sorting (FACS) to isolate the same population of CD45^+^CD11b^+^Ly6G^−^Ly6C^hi^ monocytes (Figure 2A). Following single-cell capture and RNA sequencing, principal component analysis revealed the transcriptomes from biological replicates segregated together in clusters based on experimental group (Figure 2B).

We subjected 5825 single-cell transcriptomes to Louvain clustering and *t*-SNE visualization (Figure 2C). Within the four experimental groups, naïve blood, blood 2h, PEC 2 h and PEC 6 h, there are nine distinct clusters. Across all nine clusters, 11,880 genes were significantly upregulated and 6403 genes were significantly down-regulated more than 2-fold compared to each other cluster (5% false discovery rate) (Appendix A).

We observed three distinct clusters within the naïve blood monocytes (Figure 2D). Within the blood 2 h monocytes, there were also three clusters, including two new clusters and cells which fell into cluster 1, which is shared with naïve blood (Figure 2D). Within the PEC 2 h monocytes, there was the emergence of two further distinct clusters (Figure 2D), and within the PEC 6 h monocytes, there were two further unique clusters (Figure 2D). Remarkably, there was only limited biological variation between mice within each experimental group, but significant biological variation between groups (Figure 2D). Our experiments revealed previously unknown heterogeneity within the CD11b^+^Ly6G^−^Ly6C^hi^ monocyte population. Differential gene expression revealed a pattern of rapid and divergent differentiation in the blood Ly6C^hi^ monocytes as early as two hours after zymosan challenge, but also showed that Ly6C^hi^ monocytes recruited to the peritoneum continued on a differentiation pathway characterised by differential gene expression of discrete subsets of genes (Figure 2E). Each cluster had a unique transcriptional profile, but since all the cells analysed are Ly6C^hi^ monocytes, it was not unsurprising that each cluster did not have unique gene identifiers; indeed, there was significant overlap in markers between clusters (Figure 2F).

Having demonstrated heterogeneity within the Ly6C^hi^ monocyte population, we next wanted to understand the pathways which are differentially regulated as monocytes are recruited from the blood to the peritoneum within the first 6 h following inflammatory insult. Pseudo-bulk analysis of monocyte clusters demonstrated that Ly6C^hi^ monocytes segregated by experimental group (time point) (Figure 3A). This was indicative of divergent transcriptomes during early activation. We fully characterised, at a single-cell level, the temporal gene expression profiles of Ly6C^hi^ monocytes in the first 6 h after zymosan challenge in vivo (Figure 3A). We identified biological processes associated with monocyte re-programming within the blood and the peritoneum, having performed pathway enrichment analysis. Two hours after zymosan stimulation, key pathways that are upregulated in 2 h blood monocytes compared to naïve blood monocytes with a z-score greater than 2.5 included those related to monocyte and macrophage acute response, including NO production, FcγR-mediated phagocytosis, and pathogen pattern recognition receptors. Pathways that were also enriched included those needed for cellular motility, including RhoA and Cdc42 signalling. The top-ranked enriched canonical pathway was oxidative phosphorylation (OxPhos) (Figure 3B).

Pathway analysis revealed that when compared to 2 h blood monocytes, the significantly upregulated pathways in 2 h PEC monocytes with a z-score less than 2.5 were associated with pattern recognition receptors and TREM-1 signalling (Figure 3C), and the downregulated pathways were associated with cellular motility, the actin-based motility of Rho, actin cytoskeletal signalling, and Cdc42 signalling. Pathway enrichment analysis identified OxPhos, cholesterol biosynthesis, and glycolysis to be among the top-ranked pathways in 6 h PEC monocytes vs. 2 h PEC monocytes (Figure 3D). Pseudo-bulk sequencing revealed numerous pathways to be enhanced between each of the experimental groups. OxPhos was upregulated in 2 h blood monocytes compared to naïve blood and in 6 h PEC monocytes compared to 2 h PEC monocytes. Our results demonstrate that OxPhos could be important in migration (monocytes) and differentiation (monocyte to macrophage). Heat map analysis of the top 30 differentially expressed genes from each pairwise comparison, naïve blood vs. 2 h blood (Figure 3E), 2 h blood vs. 2 h PEC (Figure 3F), and 2 h PEC vs. 6 h PEC (Figure 3G), revealed distinctive patterns of differential gene expression whereby not all the individual cells from each experimental group followed the same expression pattern. Our single-cell transcriptional profiling strongly suggests significantly more complexity within the system than could be appreciated by looking at the bulk differential expression between experimental groups.

### 3.3. A Sub-Set of M2-Like Ly6C^hi^ Monocytes Upregulates OxPhos

Pathway enrichment analysis using IPA software revealed that cells in Cluster 0 have enrichment for genes related to coronavirus pathogenesis, HIFα, and interferon signalling (Figure 4A,B). Network analysis was performed on the top 250 differentially expressed genes from Cluster 0 vs. all other cells, confirming the key involvement of genes known to be highly expressed in M1 macrophages, including key transcription factors STAT4 and IRF7 (Figure 4C). This suggests that these are a pro-inflammatory subset of Ly6C^hi^ monocytes.

Pathway enrichment analysis predicted that cells in cluster 2 have positive signatures for pathways such as OxPhos, fMLP signalling, and the NFAT regulation of immune responses (Figure 4D,E). Network analysis was performed on the top 250 differentially expressed genes from Cluster 2 vs. all other cells, confirming the key involvement of genes known to be highly expressed in M2 macrophages, including Arg1 and IL4R (Figure 4F). These pathways are known to be important for M2 macrophages differentiation and regulation of myeloid anti-inflammatory functions and inflammation resolution. These analyses identify that as early as 2 h after an inflammatory stimulus, Ly6C^hi^ monocytes are pre-programmed in the blood to have a distinct cellular and functional fate at the site of inflammation.

### 3.4. Early Changes in the Single-Cell Transcriptome Predict Heterogeneity and Divergence in Ly6C^hi^ Monocyte Differentiation

To better understand the origin of the CD11b^+^Ly6C^hi^ monocyte populations present in the peritoneal cavity at 6 h, we used Monocle 3, which can predict pseudo-time cell trajectories (Figure 5A). This analysis segregated cells based on the experimental group the cells were from, e.g., naïve blood or the in vivo compartment (blood vs. peritoneum) (Figure 5A).

The first analysis demonstrated that there was one trajectory within the blood (named Trajectory 1), which has its origin in cluster 1, extends to cells present in the 2 h blood samples, and projects with a strong pseudo-time prediction score to cluster 4. Cluster 4 is exclusively present in 2 h PEC cells. (Figure 5B). Strikingly, these cells do not follow any further differentiation beyond 2 h; cluster 4 cells are also not present within the peritoneum by the 6 h time point.

Cells from Trajectory 2 are then re-clustered, with the root cells being 2 h blood, resulting in the trajectory being subdivided into two discrete trajectories. Sub-trajectory 2a was a linear progression of cells from cluster 5 present in the 2 h blood to become cells present in cluster 3 in the 2 h PEC group and ended in cluster 0 present in the 6 h PEC group (Figure 5C). Sub-trajectory 2b is a direct progression of cells from cluster 7 present in the 2 h blood to become cells in cluster 2 present in the 6 h PEC monocytes following zymosan challenge (Figure 5C). Taken together, these data support that idea that macrophage fate at the site of inflammation is pre-determined.

### 3.5. Oxidative Phosphorylation Is Required for Monocyte/Macrophage Chemotaxis

The top enriched pathways between naïve blood and 2 h blood monocytes was OxPhos, suggesting that OxPhos is the mechanism by which naïve blood monocytes generate the ATP needed for migration to the site of inflammation. To test this hypothesis, we utilised an ex vivo real time chemotaxis system to investigate the role of OxPhos in monocyte/macrophage chemotaxis.

Firstly, we wanted to inhibit ATP production from OxPhos and determine if this affected monocyte chemotaxis. We have previously shown that BioGel-elicited monocyte/macrophages can undergo chemotaxis towards CCL2, while bone-marrow-derived macrophages (BMDM) undergo chemotaxis towards C5a. Oligomycin inhibits the ATPase F0/F1, which blocks ATP production via the electron transport chain (Figure 6A). Pre-treatment with oligomycin (1 µM; 15 min prior to chemotaxis) significantly inhibited BioGel-elicited monocyte/macrophage chemotaxis towards CCL2 (Figure 6B), assessed according to the Cell Index Max–Min and area under the curve. Similar results were obtained by using inhibitors of complexes I, II, III, and IV (Figure 6C–F), suggesting that ATP production from an intact electron transport chain via OxPhos is critical for monocyte chemotaxis. We confirmed that OxPhos is also needed for BMDM chemotaxis towards C5a. Pre-treatment with oligomycin (1 µM; 15 min prior to chemotaxis) significantly inhibited BMDM chemotaxis towards C5a (Figure 6G). This experiment demonstrates that murine monocytes and murine BMDMs have the same requirement for ATP generation from OxPhos to undergo chemotaxis. ATP can also be generated directly via glycolysis or indirectly via fatty acid oxidation (FAO). Therefore, we next pre-treated BMDMs with 2-deoxyglucose (to inhibit glycolysis) and etomoxir (to inhibit FAO). Importantly, cells pre-treated with 2-deoxyglucose (Figure 6H) or etomoxir (Figure 6I) displayed no reduction in capacity to undergo chemotaxis.

### 3.6. Oxidative Phosphorylation Is Required for Human Monocyte Chemotaxis

We next wanted to confirm the results we obtained in primary murine monocytes and macrophages in primary human monocytes. Human monocytes were immunomagnetically sorted from peripheral blood mono-nuclear cells isolated from leukocytes cones. Primary human monocytes were pre-treated with the ATPase F0/F1 inhibitor oligomycin (1 µM) for 15 min prior to being added to the top chamber of the CIM-16 plate. Inhibiting ATP production via OxPhos inhibited primary human monocytes’ ability to undergo chemotaxis towards CCL2 (Figure 6J), analysed using both the maximal rate of chemotaxis (Max–Min CI) and the total migration (AUC) (Figure 6J), clearly demonstrating that monocyte/macrophages use ATP generated from an intact electron transport chain to power chemotaxis.

### 3.7. Oxidative Phosphorylation Is Required for Murine Monocyte-to-Macrophage Differentiation

Having demonstrated that energy metabolism via OxPhos is one of the top ranked canonical pathways to be switched on following inflammatory stimulus, we wanted to test whether OxPhos was needed for monocyte-to-macrophage differentiation. Monocytes were isolated from mouse bone marrow using magnetic separation and cultured for 7 days in L-929 conditioned media (containing M-CSF), and to differentiate between BMDMs and cells, they were analysed via flow cytometry using the cell surface markers CD11b and F4/80 (Figure 7A). To test if ATP production from OxPhos is needed for monocyte-to-macrophage differentiation, monocytes were treated with 1 µM oligomycin or a vehicle during the 7-day differentiation protocol. Treatment with oligomycin significantly reduced the formation of CD11b^+^F4/80^+^ macrophages, confirming that ATP production from OxPhos is essential for monocyte-to-macrophage differentiation (Figure 7B), and there was no decrease in cellular reducing capacity/viability (Figure 7C). These data suggest that the cells are viable but critically need OxPhos, from an intact electron transport chain to be able to differentiate into mature macrophages.

### 3.8. Oxidative Phosphorylation Is Required for Murine M(IL-4)/Resolution Macrophage Differentiation

At sites of inflammation, macrophages are thought to polarise to either M1, which leads to continued inflammation, or to M2, which is associated with inflammation resolution and the maintenance of tissue homeostasis and repair. We therefore polarised BMDMs to M(LPS + INFy) and M(IL-4) and looked at bioenergetics using the Seahorse Analyser. M0 macrophages are quiescent at the basal state and are able to utilise OxPhos. We confirmed that M1 macrophages do not utilise OxPhos (Figure 7D,E), while M(IL-4) BMDMs have an enhanced basal oxygen consumption rate (OCR) and spare capacity OCR (Figure 7D,E). To test whether OxPhos is needed for macrophage M(IL-4) polarisation, BMDMs were polarised to M(IL-4) in the presence and absence of an inhibitor of OxPhos. Inhibiting ATP production via OxPhos using oligomycin reduced M(IL-4) polarisation. Flow cytometry analysis of cell surface markers revealed that oligomycin reduced the expression of IL-4-induced CD206 and CD71 markers of alternative macrophage activation (Figure 7F,G). This result shows that M2/M(IL-4) macrophage differentiation is heavily dependent on OxPhos, consistent with the strong upregulation of genes associated with OxPhos seen in our single-cell RNA Seq experiments (Figure 3).

### 3.9. Oxidative Phosphorylation Is Required for Human Monocyte-Derived Macrophage M(IL-4) Polarisation

Isolated human monocytes were grown for 7 days in the presence of h-MCSF to generate human monocyte-derived macrophages (hMoDMs) and polarised on day 6 to either a M(LPS + INFy) and M(IL-4) macrophage for 18 h. Similar to the results obtained using murine M(IL-4) hMoDMs, human M(IL-4) hMoDMs have an enhanced ability to utilise OxPhos compared to M0 or M(LPS + INFy) hMoDMs (Figure 7H,I). Indeed, when M(IL-4) hMoDMs are polarised in the presence of oligomycin, they fail to fully polarise into M(IL-4) hMoDMs displaying significantly lower levels of CD206^+^ compared to M(IL-4) hMoDMs treated with a vehicle (Figure 7J,K). These results are consistent with our murine transcriptional in vivo data, suggesting a key role for OxPhos in resolution-phase M2-like macrophage development.

## 4. Discussion

In the present study, we sought to better understand at a single-cell level the earliest transcriptional events determining blood monocyte fate as they are recruited to and differentiate within a site of resolving inflammation. Our experiments reveal there is significant heterogeneity within the circulating Ly6C^hi^ monocyte populations, and that Ly6C^hi^ monocytes undergo rapid transcriptional reprogramming in the blood before entering the site of ongoing inflammation. Critically, we have demonstrated that there is a rapid and dynamic metabolic shift toward oxidative metabolism. This metabolic shift occurs earlier than has been demonstrated before, specifically prior to monocytes displaying classical macrophage markers, which are maintained as they differentiate. Indeed, we demonstrate for the first time a requirement for ATP production from the intact electron transport chain for both monocyte/macrophage chemotaxis and monocyte-to-macrophage differentiation and M(IL-4) polarisation.

Under a steady state, it is widely accepted that tissue-resident macrophages self-maintain throughout their adult life, with minimal contribution from circulating monocytes [12,39,40]. However, this model has been challenged. Most tissues are now recognized to contain multiple macrophage populations localised to their distinct microanatomical domains [41,42]. Each of these myeloid cell populations differs in its ontogeny and capacity for self-renewal, and each macrophage population likely plays a specialised role in tissue homeostasis, tissue injury, and tissue repair [43,44]. Critically, their rate of replacement by monocyte-derived cells varies widely under a steady state. Until now, current methods have also had limited success in tracking monocyte progeny in inflamed tissues, and it was still unclear whether monocyte-derived macrophages transiently infiltrate inflamed tissues or persist locally once the inflammation resolves [39,45].

Unbiased tSNE visualisation based on Louvain clustering of single-cell RNA sequencing data revealed there is significant heterogeneity within inflammatory monocytes (CD45^+^CD11b^+^Ly6G^−^Ly6C^hi^) isolated from our four experimental conditions. Conventional cell surface expression identifies our input cells as a homogenous population; therefore, it is striking that we can identify significant heterogeneity within the cells present not only between the four experimental groups but within each experimental group. It is now widely accepted that analysis of cell surface markers vastly underestimates the heterogeneity within monocyte populations and that while cell surface expression may remain the same under inflammatory conditions, the transcriptome of individual cells may change rapidly to meet the changing demands of the cell [46,47]. Our data are consistent with a model of heightened heterogeneity within classically defined immune cell populations, and that transcriptomes may diverge before classical cell surface markers do. This highlights the need for more advanced methods beyond just following cells surface markers to determine functional diversity amongst different cell types [48,49].

Our data are consistent with a model in which differentiation into a monocyte-derived macrophage is pre-programmed prior to receiving an inflammatory stimulus at the site of inflammation. This finding does not agree with the current convention that monocytes differentiate into macrophages and then polarise following a subsequent activating signal. Our data also corroborate a recent study which identified at least two unique tissue-resident interstitial macrophages in the steady-state lung that could be distinguished by unique transcriptional profiles and spatially localized to the interstitium of the bronchovascular bundles, but not alveolar walls. In line with our findings, it was identified that other tissue-resident macrophages are blood monocyte-derived at both a steady state and following inflammatory stimulation [41,50]. Others have shown using transcriptome-based network analysis a spectrum model of human macrophage activation, i.e., macrophages in vivo do not fully fall into the classical M1/M2 paradigm [13,51]. Our data add compelling evidence that post-inflammatory tissue niches can be repopulated by a diverse range of functionally distinct Ly6C^hi^ monocyte-derived macrophages, and that these may be identified by their metabolic signature [52].

We have shown that Ly6C^hi^ monocytes rapidly upregulate key genes in the OxPhos pathway, within the first 2 h in blood monocytes. This observation suggested a need for increased ATP production during the acute activation phase in the blood prior to mobilisation to the site of inflammation. We therefore sought to determine whether ATP production from OxPhos is needed for monocyte chemotaxis to a range of physiological monocytes/macrophage chemokines using a real time chemotaxis platform [37,53]. Remodelling of the cytoskeleton to enable cell motility consists of actin polymerisation and actomyosin contraction that is sustained through ATP and GTP hydrolysis [54]. Indeed, we were able to demonstrate that when the ATPase F0/F1 was inhibited with oligomycin, this reduced the bioavailability of ATP, and there was significantly reduced murine monocyte chemotaxis towards CCL2/C5a. A similar result was obtained in human monocytes. We therefore extended our investigation to determine if disruption of the electron transport chain at different points could also alter monocytes chemotaxis, and found that the disruption of complexes I, II, III, and IV using chemical inhibitors also resulted in a significantly reduced ability of monocytes to undergo chemotaxis. Furthermore, we also showed that macrophages do not use ATP generation from glycolysis or fatty acid oxidation for chemotaxis. This contrasts with other leukocytes, such as T-cells, which use glycolysis [9]. Kim et al. also reported that redox regulation of MAPK phosphate 1 is important in monocyte migration [55]. Our data are the first to suggest a requirement of ATP production from an intact electron transport chain/OxPhos for monocyte chemotaxis.

Our transcriptomic datasets further revealed a transitional monocyte stage with a gene signature indicating active protein synthesis and turnover. The transition from quiescence (naïve blood) to cellular differentiation (2 h PEC) is dependent on increased ribosome biogenesis and protein synthesis [56] before ribosomal genes are suppressed during terminal differentiation (6 h PEC) [57]. We confirmed this experimentally, showing that a mixed bone marrow population which included hemopoietic stem cells and monocytes failed to fully differentiate into macrophages (CD11b^+^F4/80^+^) in the presence of oligomycin, reinforcing the idea that there is a critical need for OxPhos for energy-dependent processes like protein synthesis. Pathway enrichment analysis of pseudo-bulk sequencing between experimental groups revealed a central role for OxPhos at two critical time points, 2 h blood monocytes and naïve monocytes, and between 2 h PEC Mono and 6 h PEC Mono. In our cluster analysis, we see that globally, OxPhos is not upregulated in all monocyte clusters in each of the experimental groups, but rather is restricted to monocytes present in sub-trajectory 2b (Figure 3D). Only by using a single-cell RNA-transcriptomics approach can the subtleties and inherent heterogeneity of the mononuclear phagocyte system be fully appreciated.

In conclusion, we have demonstrated that there is significant previously unrecognised heterogeneity in Ly6C^hi^ monocytes. We have also revealed a requirement for rapid metabolic reprogramming in a subset of Ly6C^hi^ monocytes in the blood within 2 h, which allows them to commit along a differentiation trajectory towards an M2-like phenotype. Our work opens up the possibility to have targeted prophylactic therapies that alter monocyte metabolic capacity which facilitate enhanced macrophage M2-like polarisation to aid inflammation resolution and tissue repair if needed.

## Figures and Tables

**Figure 1 cells-13-00916-f001:**
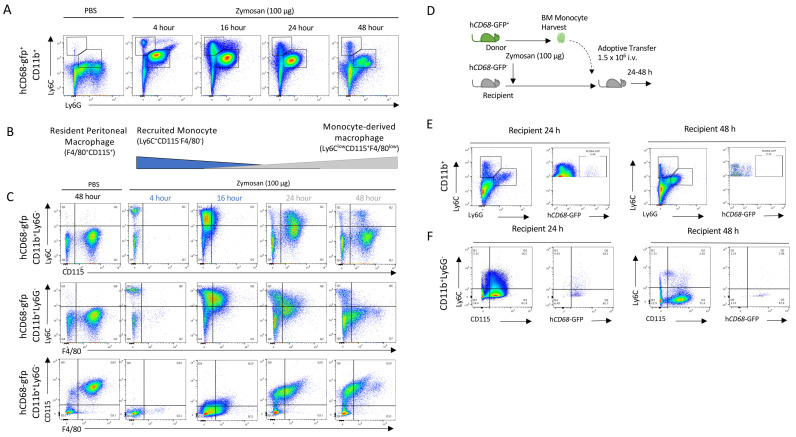
Monocytes are recruited to the peritoneum and differentiate into the monocyte-derived macrophages following zymosan challenge. (**A**) Peritoneal exudate was harvested from hCD68-gfp 4–48 h after being injected with either zymosan (100 ug; i.p.) or PBS and the cellular composition was analysed via flow cytometry. (**B**) Schematic showing the dynamic change in myeloid cell populations over time following zymosan challenge. (**C**) Representative flow cytometry of monocyte to macrophages differentiation following zymosan injection using three separate sets of cell-surface antigens, Ly6C vs. CDD115, Ly6C vs. F4/80, and CD115 vs. F4/80, after being gated on hCD68-gfp^+^. (**D**) Schematic showing adoptive transfer of bone marrow monocytes from donor hCD68-gfp^+^ mice into recipient hCD68-gfp^−^ following zymosan (100 ug; i.p.). (**E**) Representative flow cytometry of peritoneal exudates 24 h and 48 h after adoptive transfer of hCD68-gfp^+^ bone marrow monocytes showing recruited monocytes (CD11b^+^Ly6G^−^Ly6C^+^) and neutrophils (CD11b^+^Ly6G^+^Ly6C^+^) and hCD68-gfp expression. (**F**) Representative flow cytometry of peritoneal exudates 24 h and 48 h after adoptive transfer of hCD68-gfp^+^ bone marrow monocytes showing monocyte to macrophages differentiation and hCD68-gfp expression.

**Figure 2 cells-13-00916-f002:**
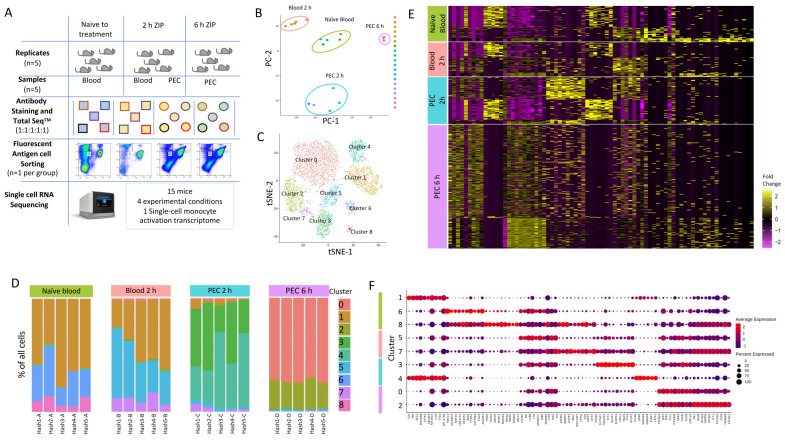
Single-cell RNA sequencing to profile Ly6C^hi^ monocytes following zymosan-induced peritonitis reveals significant heterogeneity. (**A**) Workflow and experimental scheme including experimental conditions (naïve to treatment, 2 h of zymosan-induced peritonitis (ZIP), and 6 h of ZIP), cell types harvested (blood or Peritoneal exudates), cells sorted (CD11b^+^Ly6G^−^Ly6C^+^), and processing workflow (cell hashing and 10x Genomics followed by Illumina High Seq 4000). (**B**) Principal component analysis (PCA) plot of bulk scRNA-seq profiles for each sample. (**C**) *t*-SNE visualization of 5825 cells following exclusion criteria, coloured by cluster identity from Louvain clustering. (**D**) Bar chart showing the percentage of cells per cluster identified based on cell hashing identity. (**E**) Heatmap of differential gene expression of cells vs. all other cells and grouped based on experimental group (naïve blood (green), 2 h blood (salmon), 2 h PEC (turquoise), and 6 h PEC (lilac). (**F**) Dot plot showing the expression of top 10 genes per cluster.

**Figure 3 cells-13-00916-f003:**
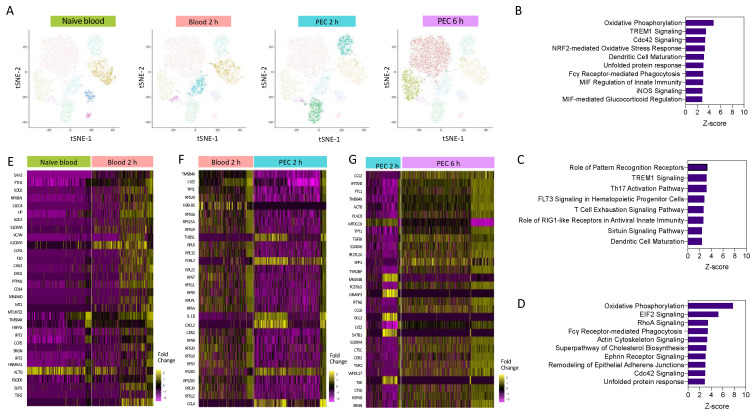
Pathways enrichment analysis reveals a central role for oxidative phosphorylation. (**A**) I-SNE visualisation of cells based on cluster, and highlighted cells present in each experimental condition. Pathway enrichment analysis between (**B**) 2 h blood and naïve blood, (**C**) 2 h PEC and 2 h blood, and (**D**) 6 h PEC and 2 h PEC. Heat map of the top 30 differentially expressed genes between (**E**) 2 h blood and naïve blood, (**F**) 2 h PEC and 2 h blood, and (**G**) 6 h PEC and 2 h PEC.

**Figure 4 cells-13-00916-f004:**
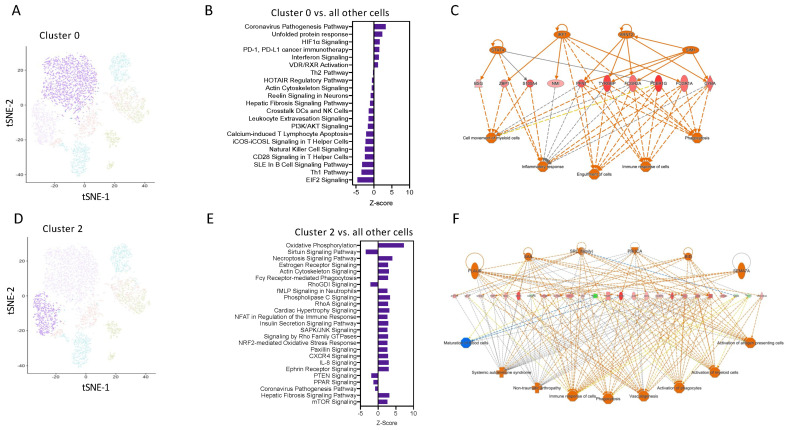
Transcript network analysis reveals that monocytes are programmed to have a pro-inflammatory or resolving phenotype as early as 6 h following zymosan challenge. (**A**) UMAP visualization of cluster 0. (**B**) Pathway enrichment analysis of cells in cluster 0 vs. all other cells. (**C**) Network analysis of the top 250 differentially expressed gene in cluster 0 vs. all other cells. (**D**) UMAP visualization of cluster 2. (**E**) Pathway enrichment analysis of cells in cluster 2 vs. all other cells. (**F**) Network analysis of the top 250 differentially expressed gene in cluster 0 vs. all other cells.

**Figure 5 cells-13-00916-f005:**
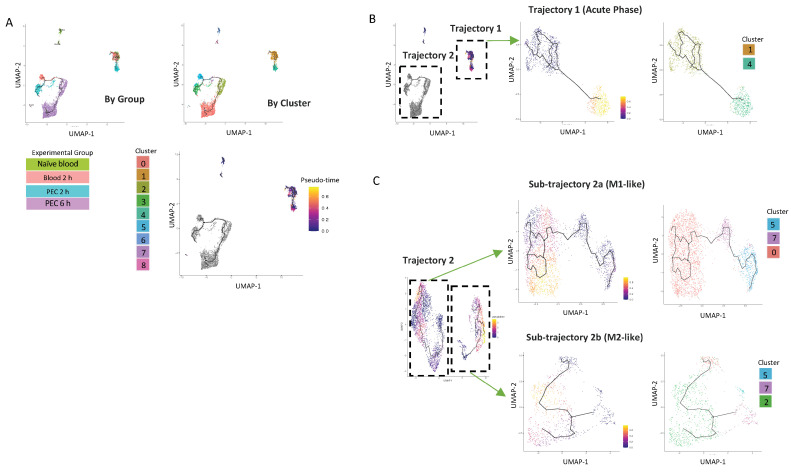
UMAP visualisation of Ly6C^hi^ monocyte differentiation trajectories in the first 6 h following zymosan challenge. (**A**) UMAP visualization annotation based on groups and clusters and by pseudotime. Iteratively, we reanalysed each of the two major trajectories shown in (**B**). Reanalysis of Trajectory 2 revealed two sub-trajectories which were each re-analysed (lower right). Results of re-analyses are shown in pseudo-time and by cluster (**C**).

**Figure 6 cells-13-00916-f006:**
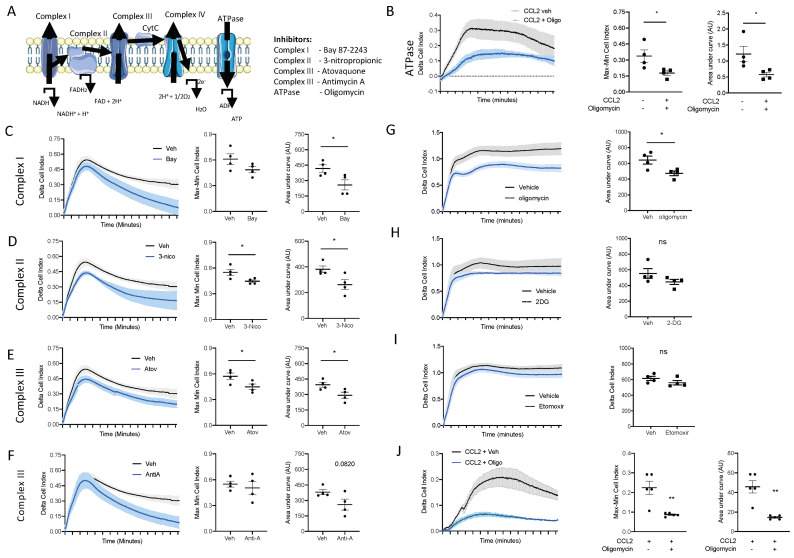
ATP production from oxidative phosphorylation is essential for monocyte and macrophage chemotaxis. (**A**) Schematic of the mammalian electron transport chain. BioGel-elicited myeloid cells were incubated with (**B**) oligomycin (1 µM), (**C**) Bay87-2243 (100 nM), (**D**) 3-nitropropionic (1 mM), (**E**) atovaquine (5 µM), or (**F**) antimycin A (0.5 µM) and a vehicle for 15 min before being added to the upper chamber (4 × 10^5^/well) of a CIM-16 plate and allowed to migrate for 6 h toward murine CCL2 (10 nM). The cell index was measured at 30 s intervals for 6 h and quantified through Max–Min or area under the curve (AUC) analysis. BMDMs were incubated with (**G**) oligomycin (1µM), (**H**) 2-deoxyglucose (50 mM), or (**I**) etomixir (3µM) or a vehicle for 15 min before being added to the upper chamber (1 × 10^5^/well) of a CIM-16 plate and allowed to migrate for 6 h toward murine C5a (10 nM). (**J**) Human monocytes isolated from buffy coats. A total of 4 × 10^5^/well were incubated with oligomycin (1 µM) or a vehicle for 15 min before being added to the upper chamber of a CIM-16 plate and allowed to migrate for 8 h toward human CCL2 (10 nM). Migration was measured with Max–Min analysis and area under the curve (AUC) analysis. Statistical analysis was conducted via one-way ANOVA with Dunnett’s multiple comparison post-test. * *p* = 0.05, ** *p* = 0.01.

**Figure 7 cells-13-00916-f007:**
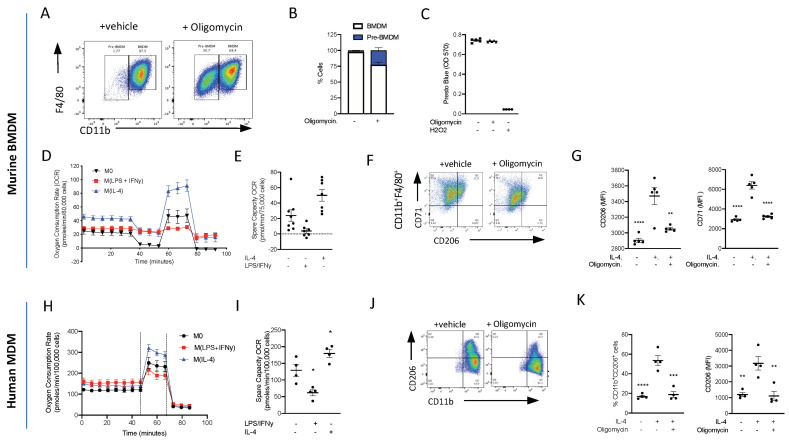
ATP production from oxidative phosphorylation is essential for monocyte-to-macrophage differentiation and M(IL-4) polarisation. (**A**) Bone marrow monocytes were isolated using immunomagnetic sorting and differentiated into BMDMs using an L-cell conditioned medium for 7 days. Representative flow cytometry of BMDMs treated with oligomycin (1 µM) or vehicle. (**B**) The percentage of CD11b^hi^F4/80^+^ and CD11b^lo^F4/80^+^ macrophages was quantified. (**C**) Viability assay of BMs grown for 7 days in L-cell conditioned medium with oligomycin (1µM) or vehicle; H_2_0_2_ was used as a positive control to induce cell death. (**D**) Extracellular flux was measured on a Seahorse XFe96 analyser. BMDMs (7.5 × 10^5^) were treated with LPS (100 ng/mL)/IFNy (20 ng/mL) or IL-4 (20 mg/mL) 18 h prior to analysis on day 7 and the oxygen consumption rate (OCR) was measured during a Mitostress test. (**E**) The spare capacity oxygen consumption rate (OCR) was quantified. (**F**) Representative flow cytometry plots of BMDMs treated with oligomycin (1 µM) or vehicle 15 min prior to stimulation with IL-4 (20 ng/mL) for 18 h. (**G**) The mean fluorescent intensity (MFI) of CD206 and CD71 was quantified. (**H**) Extracellular flux was measured on a Seahorse XFe96 analyser. Human monocytes (10 × 10^5^) were differentiated to human monocyte-derived macrophages (hMoDMs) were treated with LPS (100 ng/mL)/IFNy (20 ng/mL) or IL-4 (20 mg/mL) 18 h prior to analysis on day 7 and the oxygen consumption rate (OCR) was measured during a Mitostress test. (**I**) The spare capacity Oxygen consumption rate (OCR) was quantified. (**J**) Representative flow cytometry plots of hMoDMs treated with oligomycin (1 µM) or vehicle 15 min prior to stimulation with IL-4 (20 ng/mL) for 18 h. (**K**) The percentage of CD11b^+^CD206^+^ cells were measured and mean fluorescent intensity (MFI) of CD206 were quantified. Statistical analysis was conducted via one-way ANOVA with Dunnett’s multiple comparison post-test. * *p* = 0.05, ** *p* = 0.01, *** *p* = 0.001, **** *p* = 0.0001.

## Data Availability

All primary data is available upon reasonable request.

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
