# Peer review of "Ly6Chi Monocytes Are Metabolically Reprogrammed in the Blood during Inflammatory Stimulation and Require Intact OxPhos for Chemotaxis and Monocyte to Macrophage Differentiation"

_cells, 2024, doi:10.3390/cells13110916_

Round 1

Reviewer 1 Report

Comments and Suggestions for Authors

In this study, Purvis et al. provided evidence that Ly6Chi monocytes are heterogenous within the blood through single cell transcriptomics and that major transcriptional reprogramming events occur during the mobilization and recruitment phase. They further showed that oxidative phosphorylation (OxPhos) was one of the most important processes that take place during the changes that occur when monocytes transit from the blood to the peritoneum. In addition, they experimentally validated that OxPhos is necessary for chemotaxis, monocyte to macrophage differentiation and macrophage IL-4 polarisation. The findings are interesting but the reviewer has concerns listed below that needs to be addressed.

1.     The authors suggested that their findings depicting heterogeneity in the Ly6Chi population was “previously unknown”. They also postulated that their study of how monocytes may be pre-progammed prior to receiving an inflammatory stimulus at the site of inflammation does not agree with current convention. The reviewer noted that there are other findings prior to this study that have demonstrated heterogeneity through single cell sequencing in Ly6Chi monocyte populations (https://doi.org/10.3389/fimmu.2021.636720) and that their theory of pre-programming has also been published by another group (https://doi.org/10.1182/blood.2023020257). It is important for the authors to do a thorough literature search on studies that have performed similar investigations and cite these publications for their discussions. In view of these studies, the authors should tone down their claims of novelty.

2.     The authors discussed in lines 566 to 569 “Our data is consistent with a model in which differentiation into a monocyte-derived macrophage is pre-programmed prior to receiving an inflammatory stimulus at the site of inflammation. This finding does not agree with the current convention that monocytes differentiate into macrophages, and then polarise following a subsequent activating signal.”  Yet in their in vitro studies, the authors first differentiated their monocytes into macrophages before giving them a stimuli like LPS or IL-4. Can the authors clarify on this?

3.     Zymosan classically leads to M1 macrophages. Yet in this study that uses low dose zymosan for their transcriptomic data, the authors suggested that Oxphos was needed for the differentiation of M2 and not M1 macrophages. There seems to be a misalignment in the experimental setup and mechanistic conclusions in this aspect. Can the authors provide an explanation for this and conduct in vitro experiments using zymosan to determine how Oxphos is required in this case? This is especially important since the authors postulated in their discussion that the community should define macrophages by their metabolic signature yet it is increasingly apparent that the M1/M2 dichotomy does not always happen in vivo.

4.     The authors demonstrated in Figure 5 that Oxphos is needed for differentiation of macrophages through expression of CD11b. This is not sufficient to conclude that differentiation is impeded. The authors should perform cytospins and H&E stainings of cells that have been treated with Oligomycin and show that they look morphologically different or provide a functional assay such as phagocytosis to show differences in differentiation.

Author Response

A file containing a point-by-point response to the reviewers comments has been uploaded.

Reviewer 2 Report

Comments and Suggestions for Authors

In the present manuscript (Cells-3012229), using single cell RNA seq analysis, Purvis et al., highlighted a previously unrecognized heterogeneity within inflammatory monocytes (Ly6Chi monocytes) isolated in different conditions (blood, peritoneal exudal cell monocytes) upon zymosan-induced peritonitis in mice. They also revealed a requirement for a rapid metabolic reprogramming in a subset of Ly6Chi monocytes in the blood (within 2 hours) which allow their priming for further differentiation and  polarization towards a M2-like phenotype. From a translational point of view this works open the possibility to manipulate M2-like polarization of macrophages to resolve inflammation and to trigger tissue repair.

This is a well-written and interesting manuscript. Overall, the conclusions drawn by the authors are supported by the presented data. The following points should however be addressed.

Major point

In the first part of the manuscript, in vivo experiments yielded some very interesting single cell RNA seq data set, identifying a high degree of heterogeneity in Ly6Chi monocytes and highlighting the importance of oxidative phosphorylation in generating the ATP required for the differentiation and subsequent polarization of these monocytes to macrophages during zymosan-induced peritonitis in mice. Next, additional experiments to demonstrate this, use ex vivo human or murine macrophages incubated with different mitochondrial complexes inhibitors, identifying the inhibition of M2-like macrophage polarization. It is however expected that single cell RNA seq experiments should also provide robust information on which monocytes/macrophages subsets (M2-like for instance) are present in situ in mice, the cytokines they produced (pro or anti-inflammatory) and whether they are overexpressing M2-like markers. The authors should at least address this point in the discussion section.

More minor points

There are some inconsistencies throughout the manuscript that should edited/corrected.

1) Ideally, supplementary Figure 1 should be described at the beginning of the Results section rather than in the Materials and Methods section.

2) Figure 1D is not mentioned in the Results section

3) Figure 1E is of insufficient quality (not complete)

4) Figure 1G is mentioned in the text but there is no Figure 1G linked to Figure 1

5) Figure 2: For convenience, please indicate the number of each cluster in the histograms of Figure 2D, since some colors are not easily distinguishable on this figure.

6) The text relative to Figure 4J is lacking in the Results section

7) Lane 431: Figure 6G should be Figure 4G

8) There is no description of Figure 5J in the text

9) Please explain the rational to use alternatively male or female mice depending on the experiments

Author Response

A point-by-point response to the reviewers comments has been up loaded.
